# Analysis of the Correlation between Persimmon Fruit-Sugar Components and Taste Traits from Germplasm Evaluation

**DOI:** 10.3390/ijms25147803

**Published:** 2024-07-16

**Authors:** Yi Dong, Cuiyu Liu, Bangchu Gong, Xu Yang, Kaiyun Wu, Zhihui Yue, Yang Xu

**Affiliations:** Research Institute of Subtropics Forestry, Chinese Academy of Forestry Sciences, Hangzhou 311400, China; dongyi15030257723@163.com (Y.D.); ankar_liu@163.com (C.L.); gbc666@sina.cn (B.G.); yx2119@sina.com (X.Y.); wukaiyun99@163.com (K.W.); 15662098675@163.com (Z.Y.)

**Keywords:** *Diospyros kaki*, persimmon germplasm, GC-MS, sugar composition, E-tongue, taste quality

## Abstract

Persimmon fruits are brightly colored and nutritious and are fruits that contain large amounts of sugar, vitamins, mineral elements, and phenolic substances. The aim of this study was to explore the differences in fruit-sugar components of different persimmon germplasms and their relationships with phenotypic and flavor indices through the determination of phenotypes and sugar components and through electronic-tongue indices, which provided the basis and inspiration for the selection of different sugar-accumulating types of persimmon fruits and the selection of high-sugar persimmon varieties. Our results showed that persimmon germplasm fruit-sugar components were dominated by sucrose, glucose and fructose and that the remaining sugar components were more diverse but less distributed among the various germplasm types. Based on the proportion of each sugar component in the fruit, persimmon germplasms can be categorized into sucrose-accumulating and reduced-sugar-accumulation types. Sucrose-accumulating types are dominated by sucrose, galactose, fucose and inositol, while reduced-sugar-accumulation types are dominated by glucose, fructose, mannose-6-phosphate, and xylose. The content of sugar components in the germplasm persimmon of fruits of different types and maturity periods of also differed, with significant differences in sugar components between PCNA (pollination-constant non-astringent) and PCA (pollination-constant astringent) fruits. Cluster analysis classified 81 persimmon germplasms into three clusters, including cluster I-A, with low glucose and fructose content, and cluster I-B, with medium glucose, fructose, and sucrose contents. Cluster II was high in sucrose and fructose. Cluster III had high contents of glucose and fructose and low contents of sucrose and inositol.

## 1. Introduction

Persimmon (*Diospyros kaki* Thunb.) is a widespread temperate and subtropical species of the genus Diospyros belonging to the Ebenaceae family [1]. Approximately 1000 varieties of *Diospyros kaki* have been found in China, and it is well documented that the persimmon has been cultivated in China for about 2000 years [1,2]. As a well-known ‘woody grain’, it is considered an economically important tree in China, Japan, Spain [3], Italy [4], and Brazil [5]. Among these countries, China is the world’s largest producer of persimmons in terms of area and production [6]. According to the Food and Agriculture Organization of the United Nations, persimmon-fruit production in China was approximately 3,429,438 tons in 2021, which accounted for 79.16% of the total production worldwide [7].

Persimmon can be divided into pollination-constant non-astringent (PCNA) and non-PCNA types based on the natural deastringency of mature fruits on trees and their genetic characteristics [7,8]. Non-PCNA persimmons can be divided into the following types: pollination-variant non-astringent (PVNA), pollination-variant astringent (PCA), and pollination-variant astringent (PVA) [9].

The persimmon fruit is mainly eaten fresh but can be frozen or dried [10]. Persimmon fruits can be processed into vinegar [11] and wine [12]. In addition, persimmons are often used as sweeteners in baked products [13]. The persimmon is a nutritionally beneficial fruit and contains high amounts of phenolic compounds, fiber, vitamins, minerals, and sugar [14,15]. The main sugars in ripe persimmon fruits are fructose, glucose, and sucrose [14,16,17,18], in addition to arabinose, mannose, rhamnose, galactose, and others [19,20]. Soluble sugars are important components of the flavor quality and nutritional composition of fruits [21,22]. The amount and types of sugars stored in fruits are among the major properties determining the postharvest quality of persimmon. It has been reported that it is the soluble sugars, mainly fructose, glucose, and sucrose, that determine the sweetness of fruits, with fructose being the sweetest, 1.73 and 2.34 times sweeter than sucrose and glucose, respectively [23,24], while glucose has the best taste. Sorbitol is a sweetener widely found in plant fruits in nature; it has a unique flavor and cool and refreshing taste. It is an important raw material for the production of low-sugar candy [25]. Arabitol is used in the food industry as a sweetener and in the production of human therapeutics as an anticariogenic agent [26]. Some studies have been carried out on the soluble sugars and flavor of persimmon fruits. Glew [27] and Ge [28] showed that the main soluble sugars in ripe *Diospyros oleifera* Cheng and *Diospyros lotus* L. were glucose, fructose, and sucrose. Wang [29] et al. and Bubba et al. [16] concluded that persimmon fruits contain low levels of sorbitol and sucrose and high levels of fructose and glucose. Pan Jun’s study [30] on Taiqiu and Nantongxiaofangshi showed that the sugar components in persimmon include malt hexose, cotton-seed sugar, and mannitol, in addition to the three main sugar components and that persimmon is a glucose-accumulating fruit. Cui [31] purified two neutral sugars from persimmon peels, which are mainly composed of arabinose, galactose, glucose, xylose, and mannose.

Sugars are substances that influence fruit flavor and quality indicators and are the basic raw materials for the synthesis of many nutrients such as pigments and amino acids. They also act as signaling molecules and influence metabolic processes [32]. Therefore, the study of sugar components can not only solve the problems related to fruit flavor and quality but can also be an important basis for genetic improvement and the selection of high-sugar varieties. Sugars, as complex compounds, are difficult to detect and separate. Gas chromatography coupled with mass spectrometry (GC-MS) combines the high separation and identification capabilities of both techniques. It has a relatively comprehensive range of coverage [33] and has been refined as an experimental method for the identification of sugar components in fruits [34,35]. 

The phenotypic traits of fruit, such as size, color, and shape, are important considerations for consumers [36]. Flavor is one of the main reasons why consumers buy fruit [37]. Consumers have an important role in determining fruit quality and base their judgment on appearance and taste [38]. Hard-ripening persimmons have a ‘crunchy’, firm texture, which increases consumers’ willingness to buy. Taste is an important part of the quality of persimmon fruits and thus plays a key role in the persimmon’s final presentation quality. In some studies, the taste quality of the fruit is mainly assessed by professional sensory evaluators. This traditional evaluation method has many shortcomings, and subjective factors have a significant impact, making taste difficult to establish with objective data. The electronic tongue is an objective, fast, and economical sensory-evaluation method commonly used for the study of taste components. This important technique has been widely used in the study of tea [39], flavor enhancers [40], natural medicine, and red wines [41].

Currently, research on soluble sugars in persimmon germplasm primarily focuses on compositional content and nutrient content [42]. However, reports on the effect of soluble sugars, especially trace sugars, on persimmon-fruit flavor have not been published. Furthermore, few researchers have combined data on phenotypes with data on sugars and data collected from e-tongues to explore the association of persimmon sugar components with agronomic traits and flavor. Thus, it is necessary to establish and develop the relationship between morphological traits and soluble sugars in persimmon germplasms and to comprehensively analyze the mechanism of accumulation of sugar components and taste effects in persimmon, not only for persimmon resource management but also for investigation of genetic diversity [43]. In this study, 81 persimmon germplasm resources were investigated for 18 phenotypic traits during the ripening period, and the germplasm sugar components and flavor indices were determined using the gas chromatography–mass spectrometry and electronic-tongue methods. A total of 24 soluble-sugar components were analyzed with correlation analysis and stepwise linear regression analysis to explore the relationship between phenotypic traits, flavor traits, and sugar components. The accumulation of soluble-sugar components in persimmon germplasms of different types and from different sampling periods was analyzed, and the differences in fruit-sugar components and their contributions to each flavor index in different accumulation types and clusters were clarified. The above information will provide a basis for the identification of high-quality germplasm resources, conservation, and breeding and provide a theoretical basis for efficient cultivation.

## 2. Results

### 2.1. Characterization of Phenotypic Traits in Persimmon Germplasm

In the present study, a phenotype analysis was conducted on 81 persimmon germplasms, focusing on 18 traits (Appendix A). The fruit weight, volume, length, diameter, and shape index collectively served as indicators of fruit size. Water content, soluble-solids content, and fruit hardness are recognized as crucial indicators for assessing the physicochemical properties of fruit. The fruit skin-color and pulp-color traits were categorized as color indicators. 

Table 1 presents the mean value, standard deviation, and coefficient of variation for the 18 phenotypic traits. The mean values of fruit weight and fruit volume were 101.88 and 106.14, respectively, and the corresponding coefficients of variation were 39.45% and 41.28%, close to 40%. Fruit weight and volume were highest in YL-291, with mean values of 225.57 g and 254.74 cm^3^, followed by YL-304. YL-711 had the smallest fruit weight, and Guangzhoujixinshi had the smallest fruit volume, at 21.58 g and 22.31 cm^3^, respectively. The mean value of the fruit-shape index was 0.99, ranging from 0.76 to 1.47, with a coefficient of variation of 17.44%. YL-711 had the largest fruit shape index (1.47), whereas Jiro had the smallest (0.76). Additionally, the mean value of fruit water content measured 74.06%, with the smallest coefficient of variation, at 4%. Soluble solids and fruit firmness averaged 19.47 Brix° and 8.44 kg/cm^3^, respectively, with coefficients of variation exceeding 15%. Guangzhoujixinshi had the highest soluble-solids content, and Oku-gosho had the highest hardness. Within the spectrum of fruit color indices, both fruit skin and pulp exhibited the highest coefficient of variation for a* and the lowest for L*. The coefficients of variation for fruit skin and pulp L* were 5.76% and 9.61%, respectively. The largest values of a* of the fruit skin and pulp were found in YL 334 and Z, with values of 29.93 and 14.35. The above results indicate that the phenotypic traits differed significantly among germplasms, which is a favorable condition for the selection of superior traits.

According to the statistical analysis of the phenotypic genetic-diversity index of persimmon germplasm resources (Figure 1), it can be seen that the phenotypic-genetic diversity indices of persimmon germplasm are in the range 4.25–4.39. Among the values, those of FW, FV, and Pa* were 4.32, 4.31 and 4.25, respectively.

### 2.2. Determination of Soluble Sugars in Persimmon Germplasm

#### 2.2.1. Composition of Soluble-Sugar Components in Fruits of Different Persimmon Varieties

The contents and percentages of soluble-sugar components of 81 persimmon germplasms were determined and calculated (Appendix A), and a total of 24 sugars, including 20 monosaccharides and 4 disaccharides, were detected. 2-Ace-2-Deo-D-Glucosamine was detected in only 10 germplasms; ribono-1-4-lactone, ribose-5-pho-Ba, glucuronic-A, xylitol, and xylulose were detected in a few germplasms at very low levels; the remaining 18 soluble sugars were detected in all germplasms (Table 2). 

The composition of soluble sugars was consistent across persimmon germplasms, whereas the proportion of soluble sugars varied. The sugar components with high percentages in the fruit were sucrose, glucose, and fructose, followed by inositol and galactose. The vast majority of samples (76/81) were dominated by sucrose, with levels ranging from 23.3% to 84.95%. Seventy persimmon germplasms exhibited sucrose levels exceeding 50% of the total sugar and in seven samples, the levels were even greater than 80% (Oku-gosho, Kazusa, Sinami, Dashuishi, YL 286, YL 291, and YL 296). Notably, only six samples had less than 40% sucrose (Haianxiaofangshi, YL 200, YL 204, lvnaitou, and YL 261). Glucose content ranked second, varying from 8.7% to 41.78%. The percentage of glucose was significantly higher in Haianxiaofangshi and YL 200 than in other germplasms (41.24% and 41.78%). Oku-gosho and Sinami had the lowest glucose content as a percentage of total sugar, at less than 10%. The third-highest percentage was that of fructose, ranging from 5.54 to 35.23%, with Haianxiaofangshi having the highest fructose content and Sinami the lowest. T exhibited the highest inositol and galactose contents, at 1.56% and 0.37%, respectively, while other varieties had less than 1% inositol.

The persimmon germplasm can be classified into reduced-sugar-accumulation and sucrose-accumulating types. Sucrose-accumulating types accumulate sucrose mainly at maturity, with a low proportion of other sugar components. The reduced-sugar-accumulation type mainly accumulates glucose and fructose, and the sucrose is decomposed by sucrose synthase and invertase; thus, the sucrose content is low. In this study, reduced-sugar-accumulation germplasm accounted for only 11 samples, and the remaining 70 samples were sucrose-accumulating.

According to the statistical analysis of the genetic-diversity index of fruit-sugar components of persimmon germplasm (Figure 2), it is known that the genetic-diversity index of the fruit-sugar component of persimmon germplasm ranged from 0.92 to 4.38. Among them, Gal-A and 2-Ace-2-Deo-D-Glucosamine were 3.25 and 0.92, respectively.

#### 2.2.2. Content of Soluble-Sugar Components in Fruits of Different Persimmon Varieties

Heatmap analysis was conducted to assess the content of soluble-sugar components in persimmon germplasm (Figure 3). The total sugar content in persimmon fruits ranged from 470.49 to 984.55 mg/g FW, with an average of 692.61 mg/g FW. Jinhongshi, YL 200, YL 330, and YL 339 exhibited total sugar contents exceeding 900 mg/g, while germplasms YL 265, YL 362, and Fangshi had the lowest levels, below 500 mg/g.

Significant variations in sucrose, glucose, and fructose contents were observed across germplasms. The sucrose content ranged from 165.91 to 701.58 mg/g, with a mean value of 446.3 mg/g FW and a coefficient of variation of 29.35%. The four germplasms YL 286, Sinami, YL 348, and Oku-gosho had the highest sucrose contents, while Haianxiaofangshi had the lowest sucrose content. The coefficient of variation for glucose among persimmon germplasms was 38.84%, with the greatest value being 379.45 mg/g in YL 200. The fructose content exhibited maximum variability, with a coefficient of variation of 47.32%. Four germplasms had fructose levels surpassing 200 mg/g, namely, Haianxiaofangshi, Jinhongshi, YL 200, and YL 261. The lowest fructose content was 39.35 mg/g FW in Fangshi.

The content of inositol in Sinami, T, Gosho, Fuyu, and YL 325 was significantly higher than that in other varieties, the highest concentration being 9.68 mg/g FW in T. Fuyu had the highest D-galactose content at 2.72 mg/g, which is about twice the concentration in the rest of the varieties. The highest sorbitol content was found in YL 272 (1.11 mg/g FW) and the lowest in Taiwanhongshi (10.07 mg/g FW). The D-galacturonic acid content of YL 200 was 4.97 mg/g, and that of YL 272 was 3.42 mg/g, which was significantly higher than other varieties. The remaining soluble-sugar components mostly maintained levels below 1 mg/g in the majority of germplasms.

#### 2.2.3. Differences in the Contents of Soluble-Sugar Components in Different Persimmon Germplasm Groups

Based on the classification criteria of persimmon germplasm, 81 persimmon germplasms were categorized into PCNA, PVNA, and PCA types. Fruit can be divided into three categories depending on its ripening period: early-, medium-, and late-maturing. Differences in the contents of the sugar components across the different classifications of persimmon germplasm are shown in Table 3. Due to the high contents of glucose, fructose, and sucrose in the fruit, which far exceeded the contents of the other sugar components, the results in the stacked diagram would have obscured the contribution of the remaining sugar components, and therefore glucose, fructose, and sucrose were analyzed separately from the remaining sugar components (Figure 4). The total sugar content across the different classifications was basically the same, at about 700 mg/g.

PCNA germplasm contained more sucrose and inositol than did germplasm from the PVNA and PCA types, but its glucose and fructose contents were lower than those in both. Fructose and glucose contents were similar in the PVNA and PCA germplasms. The inositol content was 4.084 mg/g in PCNA germplasm, twice as much as that in the PCA type (1.799 mg/g). The xylose content of PVNA germplasm was twice that of the PCNA and PCA categories, and the contents of galactose and arabinose were significantly higher in the PVNA type than in the PCA germplasm. The mannose phosphate content was significantly higher in PCA germplasm than in that of other groups. The content of cellobiose in PCA germplasm was 0.139 mg/g, which was significantly higher than that of the PCNA category.

Sucrose and glucose content did not vary significantly in germplasms at different maturity stages, while there were significant differences in fructose and inositol contents, with the inositol content of early-maturing germplasm (3.661 mg/g) being significantly higher than that of medium and late-maturing germplasms, with the latter two groups having close to about 2 mg/g of inositol. Similarly, arabinose and rhamnose contents were significantly higher than those of medium and late-maturing germplasms. The cellobiose and d-mannose 6-phosphate disodium salt contents of the medium-maturing type were significantly higher than those of the early- and late-maturing varieties.

### 2.3. Correlation Analysis of Various Plastid Phenotypic Traits with Soluble-Sugar Components

Spearman correlation analysis of various phenotypic traits and soluble-sugar components in fruits (Appendix A) showed that 203 pairs of traits were significantly and predominantly positively correlated.

There were significant positive correlations between fruit volume, weight, transverse diameter, and longitudinal diameter (Figure 5B). In addition, weight and volume were significantly positively correlated with water content and PL* value. Fruit soluble-solids content was highly significantly correlated with water content (−0.363) and more than half of the peel-color indicators, with values ranging from 0.303 to 0.352. Significant correlations existed among most of the fruit color traits. There was a highly significant negative correlation between Ska* and SKh. SKL* was highly significantly correlated with SKb*, SKc, and PL*. In addition, there were highly significant correlations between b and c, both in fruit skin and in pulp (0.954 and 0.989).

Figure 5A, indicating total sugar content, shows a highly significant correlation with sucrose (0.692) and sorbitol (0.528) contents. Sucrose exhibited a highly significant negative correlation with glucose and fructose (−0.489 and −0.446), while glucose and fructose showed a highly significant correlation, with a correlation coefficient of 0.942. Fructose and glucose were highly significantly negatively correlated with inositol, a result juxtaposed against a substantial positive correlation with cellobiose and trehalose, with correlation coefficients around 0.6. The correlation coefficient between xylose and arabinose was 0.617. The phenotypes associated with sugar components were mainly centered on indices of physical and chemical properties, along with sucrose, glucose, fructose, inositol, and rhamnose contents.

The correlation heat map showed that the only phenotype associated with sucrose was fruit water content, which exhibited a correlation coefficient of 0.405 (Figure 6). Conversely, glucose and fructose showed highly negative significant correlations with fruit water content (−0.408 and −0.401) and were significantly positively correlated with Pch, presenting coefficients of 0.25 and 0.235. Inositol showed highly significant positive correlations with water content and firmness (0.391 and 0.321). Cellobiose and galacturonic acid were exclusively correlated with fruit firmness, both showing negative correlations. The sugar component that was highly significantly correlated with soluble solids was arabinose. Rhamnose was highly significantly negatively correlated with most of the color indices, and arabinitol was negatively correlated with pulp-color indicators. The sugar component that was correlated with fruit length was phenylglucoside.

### 2.4. Principal Component Analysis of Soluble-Sugar Components in Fruits of Persimmon Germplasm Resources

Principal component analysis was used to identify the variables with the greatest impact on the observed differences [43]. As shown in Table 4 and Table 5, the first seven principal components with eigenvalues greater than 1 explained 73.60% of the total variation. The first principal component represents a combination of D-mannose, trehalose, glucose, D-arabinose, fructose and phenylglucoside, and all indicators were positively distributed on PC1. The second principal component includes glucose, inositol, D-galactose, arabinitol, barium D-ribose-5-phosphate, and sucrose, and only glucose was negatively distributed on PC2. The eigenvalues of PC1 and PC2 are 4.918 and 4.112, with variance contributions of 20.49% and 17.13%, respectively.

PC3 mainly responds to sucrose, fucose and sorbitol, and the greater the value of PC3, the higher the content of the above sugars. PC4 mainly synthesizes the data on xylose and xylitol, and these two indices are positively distributed on PC4. PC5 mainly synthesizes the two indices of arabinitol and rhamnose. PC6 and PC7 respond to xylitol and man-6-pho, respectively, and both of them are positively divided on the principal components.

### 2.5. Cluster Analysis of Persimmon Germplasm Resources Based on Sugar Components

The dendrogram based on 24 sugar components was used to divide the accessions into three major clusters (Figure 7). From bottom to top, they are Cluster I, Cluster II, and Cluster III. Thirty-eight accessions were placed into cluster I, which was divided into two sub-clusters (I-A and I-B). Thirty accessions were placed into cluster II, and the rest of the accessions were placed into cluster III. The division of the 81 germplasms into three categories not only matches the results from SPSS but also demonstrates a certain degree of rationality, with each clustered germplasm exhibiting specific patterns.

In the three clusters, glucose, fructose, and sucrose play a crucial role and are the main sugar components determining the differentiation of different cluster groups. Cluster I-A comprised 23 accessions characterized by lower levels of glucose, fructose, and sorbitol and the highest levels of xylulose. I-B comprised 15 accessions, demonstrating moderate levels of glucose, fructose, and sucrose and the highest of inositol, galactose, and mannose. Cluster II exhibited high sucrose and fructose contents and moderate levels of glucose and sorbitol. In cluster III, 13 germplasms were grouped together due to their higher values of glucose, fructose, and trehalose, coupled with lower values of sucrose and inositol.

### 2.6. Electronic-Tongue Analysis of Persimmon Fruit Flavors

Sensory evaluations were conducted using an electronic-tongue instrument. With the exception of sourness, astringency, and aftertaste-B, the taste indicators are above the tasteless point (Appendix A) and thus can be used as valid evaluation indicators. As can be seen from Figure 8, aftertaste-A and sweetness varied little among germplasms, while freshness varied considerably.

As shown in the scatterplot (Figure 9), the distribution of bitterness of various germplasms was concentrated and ranged from −5 to 3. Umami was concentrated from 6 to 15, with Guangzhoudaniuxinshi being the lowest-ranked and having the highest bitterness value. The value of richness ranged from 0 to 3 in most germplasms, and YL 349 had the lowest richness, at 0.02, which was close to the point of tastelessness. Saltiness was lowest in Houzishi, at −2.79, and the rest of the germplasms were mainly distributed between −1.5 and 3. The sweetness was centrally distributed between 2.5 and 5.5, with the highest sweetness in Jiro and Yamafuji (5.69 and 5.62).

### 2.7. Correlation Analysis of Sugar Components and Electronic-Tongue Indicators in Different Persimmon Germplasms

The correlation analysis between the effective indicators of the e-tongue and the contents of sugar components showed (Appendix A, Figure 10) that aftertaste-A was highly significantly correlated with umami and richness, with correlation coefficients of 0.606 and −0.595, respectively. Umami was highly significantly negatively correlated with richness (−0.505), and saltiness was highly significantly correlated with sweetness, with a coefficient of −0.587.

Only 12 pairs of traits showed significant correlations between sugar composition and e-tongue indicators, and richness was not correlated with sugar components. Bitterness exhibited highly significant correlations with inositol, 2-acetamido-2-deoxy-D-glucopyranose, and mannose 6-phosphate disodium salt (0.321, 0.329, and −0.32). Bitterness was significantly negatively correlated with fructose (−0.228) and cellobiose (−0.269). Saltiness exhibited a significant correlation with sucrose (0.257). Sweetness was highly significantly correlated with 2-acetamido-2-deoxy-D-glucopyranose, with a coefficient of 0.303.

#### 2.7.1. Correlation Analysis between Different Types of Sugar Accumulation and Electron-Tongue Data for Persimmon Germplasms

In this study, persimmon fruit-sugar-accumulation types were classified into sucrose-accumulating and reduced-sugar-accumulation types, and the correlation between the sugar components and the e-tongue data varied among these types.

Only 50 pairs of significant correlations were identified in reduced-sugar-accumulation persimmon germplasm (Figure 11A). Total sugar was highly significantly correlated with glucose and fructose (0.936 and 0.818). Fructose showed highly significant correlations (0.736 and 0.8) with cellobiose and man-6-pho. Inositol was highly significantly negatively correlated with fructose and mannose 6-phosphate (−0.755 and −0.845) and significantly positively correlated with galactose (0.745). Cellobiose was highly significantly positively correlated with mannose 6-phosphate disodium salt, and xylose was highly significantly positively correlated with phenylglucoside (0.745 and 0.836). Astringent aftertaste was highly significantly negatively correlated with richness (−0.916). Astringent aftertaste was highly significantly and positively correlated with total sugar (0.752), and 2-acetamido-2-deoxy-D-glucopyranose significantly positively affected sweetness (0.661).

There were 115 pairs of significantly correlated traits between sugar components and e-tongue indicators in sucrose-accumulating germplasm (Figure 11B), and total sugar had the strongest correlation with sucrose, at 0.832. Glucose was highly significantly correlated with cellobiose and trehalose (0.553 and 0.576). Fructose was highly significantly positively correlated with cellobiose and highly significantly negatively correlated with inositol (0.627 and −0.539). Arabinitol was highly significantly positively correlated with rhamnose (0.887). Among the electronic-tongue indicators, astringent aftertaste was highly significantly correlated with umami and richness (0.581 and −0.548). There was a highly significant negative correlation between umami and richness (−0.456). Sweetness was highly and significantly correlated with trehalose (0.315). Bitterness showed a highly significant positive correlation with inositol, with a coefficient of 0.345.

#### 2.7.2. Correlation Analysis of Sugar Components in Different Clusters of Persimmon Germplasm with Electronic-Tongue Data

In this study, persimmon germplasms were categorized into three clusters based on the sugar-component contents in the fruit. Correlation analyses of the above indicators in the three clusters were carried out in order to accurately reflect the relationship between sugar components and flavors in different clusters.

In Cluster I (Figure 12), total sugar was highly significantly correlated with sucrose (0.723). The correlation between galactose and phenylglucoside was highly significant (0.816). There was a highly significant correlation between mannose and trehalose (0.794), and arabinitol was highly significantly correlated with rhamnose (0.881). Bitterness showed a highly significant correlation with umami and sweetness (−0.529 and 0.472), and astringent aftertaste showed a highly significant negative correlation with richness, and saltiness showed a negative correlation with sweetness (−0.467 and −0.604). Only arabinose showed a highly significant correlation with sweetness, with a correlation coefficient of 0.492.

In Cluster II, the total sugar was significantly correlated with sucrose (0.821). Cellobiose showed a highly significant positive correlation with glucose and fructose, with correlation coefficients of 0.638 and 0.678. Arabinose showed correlation with mannose (0.661). Rhamnose was highly significantly correlated with arabinitol, with a correlation coefficient of 0.862. There was a highly significant correlation between astringent aftertaste and umami and richness (0.591 and −0.732). Richness was highly significantly negatively correlated with umami (−0.712).

Among the persimmon germplasms clustered in Cluster III, total sugar was most strongly correlated with arabinitol and significantly negatively correlated with inositol (0.703 and −0.626). Glucose and fructose showed highly significant positive correlations with cellobiose and d-mannose 6-phosphate disodium salt. Xylitol was highly significantly positively correlated with sorbitol and xylose (0.703 and 0.769) and highly significantly negatively correlated with mannose (−0.714). Rhamnose showed highly significant positive correlations with arabinitol (0.78). Astringent aftertaste showed a highly significant positive correlation with umami, with a coefficient of 0.696. Astringent aftertaste and umami were mainly affected by ribono-1,4-lactone and barium D-ribose-5-phosphate, with astringent aftertaste being highly significantly negatively correlated with ribono-1,4-lactone (−0.77).

### 2.8. Stepwise Multiple Regression Analysis

To further determine the relationship between persimmon fruit flavor and sugar-content indicators, a regression equation was obtained by using stepwise regression analysis, with each flavor value measured using the electronic-tongue data as the dependent variable and 24 fruit-sugar-content items as independent variables.

As can be seen from Appendix A, the R^2^ values of the regression equations for astringency, freshness, and sweetness were low, while the R^2^ values of the regression equations for bitterness and sourness were 0.16 and 0.139, respectively. The results of stepwise regression analysis showed that inositol and d-mannose 6-phosphate disodium salt were the main indicators affecting the bitterness, together determining 16% of the variation in the bitterness of the persimmon. Sucrose and arabinose were the main indicators affecting the saltiness, with both playing a role in determining 13.9% of the variation in saltiness. Cellobiose was the main indicator affecting aftertaste-A, while the sugar that played a decisive role in umami was galacturonic acid sodium salt, and xylose was the main indicator affecting the sweetness.

## 3. Discussion

### 3.1. Phenotypic Diversity and Sugar Component Differences in Persimmon Germplasm Resources

The study of the genetic diversity of phenotypic traits is a key step in the identification, evaluation, and protection of germplasm resources [44,45]. Phenotypic diversity represents a combination of genetic and ecological diversity, and it is essential for maintaining the long-term survival of a species [46]. In this study, 18 phenotypic traits, including fruit size, shape, and color indicators, were determined. These traits are considered to be important indicators of consumers’ willingness to buy [47]. In this study, the range of variation for these traits was 4.00–50.79%, which is in agreement with the results of previous analyses [48,49]. FWC and SKL* had the lowest coefficient of variation, indicating that they are genetically stable. This study on genetic diversity indicates that the genetic-diversity indices among fruit phenotypic traits of persimmon germplasm resources show relatively small differences. The genetic diversity indices FW, FV, and Pa* are lower, suggesting a higher level of selection. Persimmons are commonly hexaploid, and a few are ninefold [50], and polyploidy may be the main cause of variation in fruit size and shape [51]. The color differences in persimmon fruits are mainly caused by the a* values, which is consistent with the findings of Han [48]. Therefore, persimmon fruits can be classified according to their color a* value.

The sugar contents in fruits are important determinants of fruit flavor. Differences in sugar-component contents, types, and ratios contribute significantly to the flavor of fruits [50]. The content of soluble sugars in mature fruits is a key indicator for evaluating the quality of fruits. Previous studies on the sugar components in persimmon fruits mainly focused on sucrose, glucose, and fructose [51,52], and a few have dealt with 4–6 sugar components [30,53] without digging deeper into other sugar components that may be present. In this study, a large number of sugar components were studied for the first time in 81 persimmon germplasms, and in addition to the three major sugar components mentioned above, 21 trace sugars, such as myoinositol, sorbitol, and arabinose, were also determined. This is of great significance for the formation of persimmon fruit quality and the related metabolism of sugar components. The results suggest that the composition of sugar components in persimmon germplasm was essentially the same, with sucrose, glucose, and fructose accounting for 63.26%, 21.79%, and 14.36% of the total sugar, respectively. This finding is consistent with previous results on the composition of sugar components in persimmons [29,30]. The glucose content ranged from 64.42 to 258.41 mg/g, and sucrose ranged from 165.91 to 701.58 mg/g. This result suggests that the persimmon germplasm sugar components mainly existed in the form of sucrose, followed by glucose and fructose, which is in line with the results of Candir and S.D. Senter’s studies [53,54]. The remaining sugar components account for less than 1% of the total sugar content. This does not mean that these trace sugar components are unimportant; inositol plays a very important role in cell morphology, signaling, and various physiological functions of cells [55]. Arabinose is about half as sweet as sucrose; rhamnose is 33% as sweet as sucrose; and sorbitol provides a refreshing and cool taste with less sweetness than glucose. Most of the sugar components show minor genetic-diversity differences across persimmon germplasms, with genetic-diversity indices fluctuating slightly around 4. However, Gal-A and 2-Ace-2-Deo-D-Glucosamine have the lowest genetic-diversity indices, indicating the highest potential for selection.

Fruits can be classified according to their composition of soluble-sugar components into the sucrose-accumulating type and reduced-sugar-accumulation type. In this study, 70 of the persimmon germplasms had higher sucrose content and were of the sucrose-accumulating type. Candir [56] found that the sucrose content of persimmon fruits was higher than the reduced-sugar level at the end of growth. This finding is consistent with the result that most of the persimmon germplasms in this study were sucrose-accumulating during the hard ripening period. There are also differences in the sugar-component content across different types of persimmon germplasm in that the sucrose and inositol contents of PCNA germplasm were higher than those of the PVNA and PCA types, but its glucose and fructose contents were lower than those of the two groups. This result is in agreement with Senter’s finding [55]. It indicates that the type of astringency affects the type of fruit-sugar accumulation, which is in agreement with the results of Han Weijuan and Yildiz’s studies [56,57]. The level of galacturonic acid was fourfold higher in PCA compared to PCNA germplasm. The content of cellobiose was significantly higher in PCA than in PCNA germplasm. Previous studies [56,58] have found significant differences between PCNA and PCA germplasms in persimmon sugar components.

### 3.2. Correlation Analysis of Fruit-Sugar Components with Phenotypic Traits

The phenotype directly reflects the genetic diversity of plants, which is important when screening for excellent germplasm resources [59,60]. Chen [61] evaluated the internal quality parameters of persimmons and found that soluble-solids content is an intrinsic quality attribute that directly affects fruit sweetness. Sugar, as a substance that is important during fruit development, is closely related to fruit quality [62]. Therefore, it is important to explore the correlation between fruit sugar and phenotypic traits to improve fruit quality and clarify the interaction between the two. A significant positive correlation of size indices of phenotypic traits and a significant positive correlation of soluble solids with peel-color indices were found in this study; this finding is in agreement with the reports of Rodrigues et al. [63], Kviklys et al. [64], and Candir [54].

Sugars are considered essential signaling molecules for fruit growth and development. They play a key role in fruit quality, and their carbon skeletons are often transformed into different sugar components [65]. In this study, we found that total sugar was highly significantly positively correlated with sorbitol and cellobiose in addition to sucrose. Cellobiose showed highly significant positive correlation with glucose, fructose, sorbitol, and trehalose. Glucose showed a highly significant correlation with fructose (0.956). Cellobiose is produced from cellulose in the presence of cellulase and can be further hydrolyzed to glucose [66]. Glucose and fructose are mostly decomposed from sucrose [67] and are closely related. The above conclusion is consistent with the highly significant positive correlation between cellobiose, glucose, and fructose found in this study. The correlation coefficient of fructose with mannose was 0.721. Fructose and mannose can be converted by the effect of mannose isomerase and are closely related [68]. Glucose and fructose, as the main sugar components in the fruit, were highly significantly correlated with inositol, xylose, and trehalose. These results are consistent with those of similar previous studies [69,70]. Trehalose comprises two covalently linked glucose molecules [71], so the two are significantly correlated.

The phenotypes associated with the sugar components were mainly focused on the physicochemical properties and color indices of the fruits. Soluble-solids content was highly dependent on the contents of sugar components, such as fructose and sucrose [72]. Sucrose and inositol showed a highly significant positive correlation with fruit water content, which may be related to the decrease in sucrose phosphate synthase (SPS) activity due to lower water content, which in turn leads to a decrease in sucrose content [73]. Rhamnose was significantly negatively correlated with most of the color indices, and arabinitol was negatively correlated with pulp color. Several studies [74,75,76] have shown that the soluble-sugar content of fruit is critical for carotenoid accumulation, which in turn is one of the most important factors in fruit color formation. Candir’s [54] study on persimmon fruits at different stages of development found that changes in peel color were closely related to the increase in sugar content and decrease in firmness. The accumulation of both sugar components and pigments is a gradual, synergistic process [77], so it is hypothesized that the differences in pigment-accumulation efficiency are consistent with the efficiency of sugar accumulation in the germplasm.

### 3.3. Correlations between Sugar Components and Flavor Indicators among Different Germplasms

In this study, for the first time, GC-MS combined with the electron tongue method was applied to study the relationship between sugar components and flavor of persimmon germplasm fruits. There may be taste interactions among different tastes, and different tastes produce different effects when mixed [78,79]. Studies have shown that both fresh and sweet flavors have a masking effect on bitter taste. Persimmon germplasm can be categorized into different types based on the proportion of the major sugar components. Zheng and Sugiura [80] suggested a classification of sucrose-accumulating and reduced-sugar-accumulation persimmon cultivars. Differences existed between sugar components and flavor indicators of different types of fruits, with sucrose-accumulating germplasm accounting for 70 samples in this study. This is consistent with the conclusion that persimmon fruits were predominantly sucrose-accumulating in Candir and Senter’s study [53,54]. In reduced-sugar-accumulation germplasm, total sugar was mainly affected by glucose and fructose. Mannose 6-phosphate was highly significantly correlated with fructose and inositol, and xylose was highly significantly positively correlated with phenylglucoside. Total sugar in sucrose-accumulating fruits was affected by sucrose content. Galactose showed a highly significant positive correlation with mannose, phenylglucoside, and trehalose, and arabinitol was highly significantly correlated with xylose. The results showed that glucose, fructose, and mannose 6-phosphate play important roles in reducing sugar accumulation in persimmon fruits, and sucrose and galactose are decisive in the sucrose-accumulating type [81,82]. In the sucrose-accumulating types of germplasm, sweetness was highly significantly correlated with trehalose, and bitterness was mainly influenced by inositol content, whereas the sweetness of reduced-sugar-accumulation germplasm was mainly regulated by 2-acetamido-2-deoxy-D-glucopyranose and astringent aftertaste was highly significantly correlated with total sugar. Therefore, taste indicators such as sweetness in different sugar-accumulation types are regulated by different sugar components.

From the above analysis, it is clear that different germplasm types influence the correlation between sugar components and taste indicators. In addition to germplasm types, the correlations between sugar components and flavor indicators varied among the three different clusters derived from principal component analysis and cluster analysis. Cluster I can be subdivided into two subclasses, I-A and I-B. Cluster I-A is very low in glucose and fructose; cluster III has the highest contents of both; cluster I-B has intermediate contents of glucose, fructose, and sucrose; and cluster II has high contents of sucrose and fructose. The differences in sugar components among different groups can be used as an important indicator when screening for good germplasm [83]. Correlations in the cluster groupings showed that total sugars were highly significantly positively correlated with sucrose in cluster I and cluster II, while total sugars in cluster III were mainly highly significantly positively correlated with arabinose and highly significantly negatively correlated with inositol. Cellobiose was highly significantly positively correlated with glucose and fructose in cluster II and cluster III. Mannose was positively influenced by alginate and arabinose in cluster I and cluster II, respectively, and was highly significantly negatively correlated with xylitol, mainly in cluster III. In cluster I, galactose was highly significantly positively correlated with 2-acetamido-2-deoxy-D-glucopyranose; astringent aftertaste was negatively regulated by richness; and arabinose was highly significantly positively correlated with sweetness. Astringent aftertaste was positively influenced by umami in cluster III, and astringent aftertaste and umami were mainly negatively influenced by D-ribono-1,4-lactone and barium D-ribose-5-phosphate. Astringent aftertaste in cluster II was both positively correlated with umami and negatively correlated with richness and was regulated by both, and saltiness was significantly positively correlated with sucrose and significantly negatively correlated with fructose. This finding is consistent with the results of previous studies in which soluble-sugar components helped to improve sweetness and freshness in plants [84].

### 3.4. Regression Analysis of Electronic Tongue and Sugar Components

Sugar is an important factor affecting the quality and flavor of fruits, and the soluble-sugar component is not only the most important component of sweetness [85] but also has an impact on other flavor indicators [86]. The main metabolic pathway related to bitter taste in mulberry leaves was found to be galactose metabolism by Huang et al. [87]. In this study, a stepwise regression analysis revealed that the sugar components inositol and d-mannose 6-phosphate disodium salt were the main indicators affecting persimmon bitterness. Inositol triphosphate is an important effector molecule in the downstream transduction of bitter signaling [88]. Sucrose and arabinose were the main indicators affecting saltiness. An increment in saltiness estimates was found after stimulation with sucrose [89]. The sweetness of persimmon fruit was mainly influenced by xylose. This conclusion is consistent with the results of the study on sweet oranges by Liu et al. [90]. Previous research on blueberries found that glucose and fructose were of central importance in sweetness [86,91].

## 4. Materials and Methods

### 4.1. Plant Materials

All 81 persimmon germplasms were grafted and stored during the spring of 2014–2015 in a plant nursery in Lanxi City in Zhejiang Province (29°25′ N, 119°51′ E). In addition, each germplasm contained 6 plants with a spacing of 3 m × 4 m. D. kaki fruits were harvested during October and November 2022 (Table 6). According to Chinese agricultural standards, NY/T 1309-2007 ‘Technical Code for Evaluating Germplasm Resources-persimmon’, fruit picking is carried out at commercial maturity, i.e., when nearly 60% of the persimmon fruit on the entire tree reaches an inherent orange color, the fruit is slightly hard, not softened, and the seed color has turned brown. Forty intact and disease-free fruits were harvested from each germplasm and transported to the laboratory as soon as possible. Thirty fruits were selected for the determination of phenotypic traits, such as weight, volume, and water content, and then a portion of the flesh was taken for the e-tongue analysis. The remaining 10 persimmon fruits were washed in deionized water and dried with absorbent paper. The skin and core were then removed, diced, and frozen and were finally stored at −80 °C for the subsequent determination of soluble-sugar components.

### 4.2. Determination of Phenotypic Traits

Persimmon fruits with symmetrical growth and no pests or diseases were selected, and their diameter and length (0~150 mm, 0.01 mm; Shanghai Anting Scientific Instruments Factory, Shanghai, China), weight (YP502N type, Shanghai Precision Scientific Instruments Co., Ltd., Shanghai, China), and volume were measured. The fruit shape index was calculated according to the formula ‘Fruit shape index = fruit length/fruit diameter’ [92]. Each fruit was peeled equatorially, and the hardness of the two corresponding surfaces was measured with a digital hardness tester (GY-4 digital display; Zhejiang Top Instruments, Hangzhou, China). The soluble-solids content was measured at the fruits’ equatorial position using a digital display saccharimeter (PAL-1 type; Japan ATAGO Company, Saitama, Japan).

### 4.3. Extraction and GC-MS Determination of Soluble Sugars

The biological samples were vacuum freeze-dried (CentriVap; LABCONCO Company, Kansas City, MO, USA) and ball-milled (Retsch MM400; Verder Scientific, Shanghai, China) (30 Hz, 1.5 min) to powder form, and 20 mg of the powder was weighed into the corresponding numbered centrifuge tubes. Then, 500 μL of methanol/isopropanol/water (3:3:2 V/V/V) extract was added to the samples, and the samples were vortexed for 3 min and sonicated in an ice-water bath for 30 min. A total of 20 μL of internal standard solution was added to 12.5 μL of supernatant at a concentration of 250 ug/mL, which was then subjected to nitrogen blowing and lyophilization. Then, 100 μL of pyridine methoxide ammonium salt (15 mg/mL) was added, followed by incubation at 37 °C for 2 h, the addition of 100 μL of BSTFA, and incubation at 37 °C for 30 min to obtain the derivatization solution. To the derivatization solution, 800 μL of n-hexane was added. The filtrate was filtered through a 0.22 um membrane and stored in a brown injection bottle for GC-MS analysis.

The detection equipment was an Agilent 8890-5977B high-performance gas chromatograph, and the stationary phase was a DB-5MS (30 m × 0.25 mm × 0.25um) column, with helium as the carrier gas, a split flow pattern of 5:1, and a flow rate of 1 mL/min. The column chamber was warmed up according to the heating procedure of holding at 160 °C for 1 min, then increasing to 200 °C at 6 °C per minute, then at 10 per minute to 270 °C, and finally to 320 °C at 20 °C per minute and holding for 5.5 min with a 1 μL injection. The curves for different concentrations of standards were plotted, and the correlation coefficients of the linear equations were above 0.99 (Appendix A).

### 4.4. Electronic-Tongue-Based Taste Evaluation

The skin and kernel of the persimmon were removed, and a portion of pulp was selected and cut into small pieces. Next, 40 g of the sample was weighed out; it was placed in a cooking machine, mixed with 200 mL of Wahaha pure water, and stirred in a blender (Blendtec Model 575, Orem, UT, USA) for 30 s. Centrifugation at 3000 rpm for 5 min was carried out after stirring, followed by filtration using filter paper, and the filtrate was taken for testing on the machine. The e-tongue (TS-5000 Z, Kagoku, Japan), which consisted of several sensor probes and a reference probe, was dipped into the sweet persimmon infusion and reference solution to measure the taste strength. Each sample was prepared in triplicate, and each sweet persimmon infusion was measured four times to obtain an average value.

### 4.5. Statistical Analysis

Duncan’s multiple-comparison test, one-way analysis of variance (ANOVA), Pearson’s correlation analysis, cluster analysis (CA), and principal component analysis of the samples were performed using SPSS 24.0 Statistics (SPSS Inc., Chicago, IL, USA) to identify significant differences (*p* < 0.05) among accessions. The Shannon−Weaver genetic-diversity index was calculated using Past, and the specific grading was adopted from previous studies [93,94]. The graphing was performed using Origin 2021 and GraphPad Prism 9.

## 5. Conclusions

In conclusion, 18 phenotypes, 25 sugar components, and electronic-tongue indicators of 81 hard-ripening-stage persimmon germplasms were comprehensively analyzed. We found that the phenotype, sugar-component composition, and sugar content differed significantly across germplasms and that the fruit size and color index a* could be used as important indicators of germplasm phenotypic traits. It was also observed that the fruit-sugar composition of all persimmon germplasms was basically the same, with sucrose, glucose, and fructose predominating and the contents of remaining sugar components being low. The correlation between phenotypes and sugar components mainly related to the physicochemical properties and color indices of the fruits, involving several sugar components. Germplasm resources can be subdivided into subgroups based on different types and maturity stages, with PCNA germplasm having the highest sucrose and inositol content and PCA germplasm having high levels of glucose, fructose, cellobiose, and galacturonic acid.

Persimmon germplasm was categorized into sucrose-accumulating and reduced-sugar-accumulation types according to the content and proportion of sugar components in the fruits, and the correlation between the sugar components and the e-tongue taste indices varied among the different types. The main relevant substances of sucrose-accumulating germplasm are sucrose, galactose, trehalose, and inositol, and the sugar components that play a major regulatory role in reduced-sugar-accumulation germplasm are glucose, fructose, mannose 6-phosphate disodium salt, and xylose. In addition to the above groupings, sugar composition and flavor indicators varied among the three different clusters. The three clusters can be clearly categorized according to their sucrose, glucose, and fructose contents, with cluster I-A having the lowest glucose and fructose contents and cluster II having the highest sucrose and fructose contents. Among the different clusters, the correlations were mainly focused on sucrose, glucose, fructose, cellobiose, inositol, mannose, and arabinose as the sugar components and on the taste indicators of sweetness and saltiness. These results lay the foundation for the selection of persimmon fruits containing different sugar types and also provide the basis and inspiration for the selection of high-sugar persimmon varieties for breeding, which still needs to be considered from the transcriptome perspective.

## Figures and Tables

**Figure 1 ijms-25-07803-f001:**
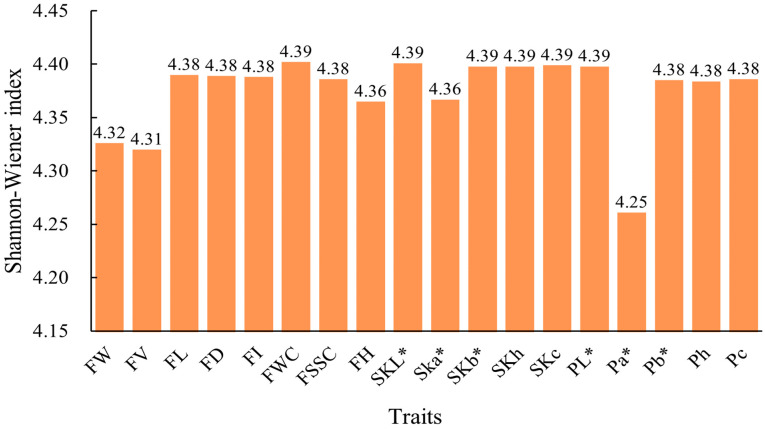
Genetic-diversity indices of 18 phenotypic traits in persimmon germplasm. Specific information can be found in Table 1. FW, fruit weight; FV, fruit volume; FL, fruit length; FD, fruit diameter; FI, fruit shape index; FWC, fruit water content; FSSC, fruit soluble solid content; FH, fruit hardness; SKL*, fruit skin lightness; Ska*, fruit skin red-green color; SKb*, fruit skin yellow-blue; SKh, fruit skin color angle; SKc, fruit skin color saturation; PL*, pulp lightness; Pa*, pulp red-green color; Pb*, pulp yellow-blue; Ph, pulp color angle; Pc, pulp color saturation.

**Figure 2 ijms-25-07803-f002:**
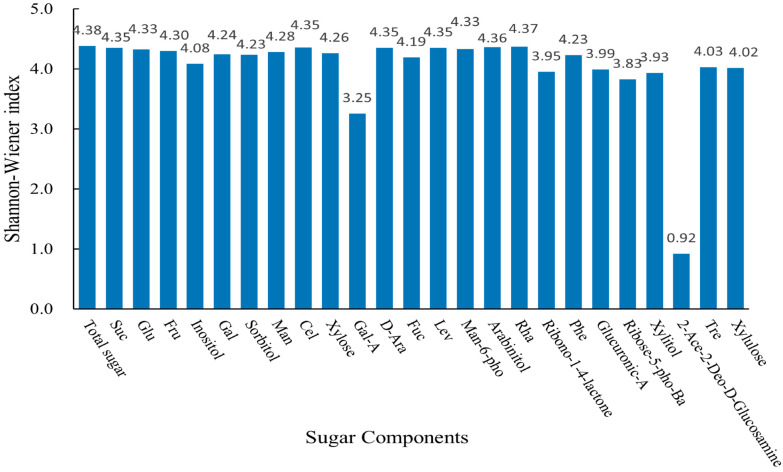
Genetic-diversity index of 25 sugar components in persimmon germplasm. Specific information can be found in Table 2.

**Figure 3 ijms-25-07803-f003:**
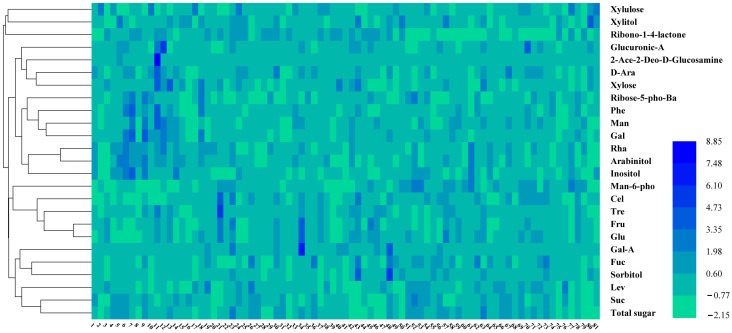
Heat map of the main sugar components cluster (blue represents high content, green represents low content; the horizontal axis represents the varieties, and the vertical axis represents the soluble-sugar components).

**Figure 4 ijms-25-07803-f004:**
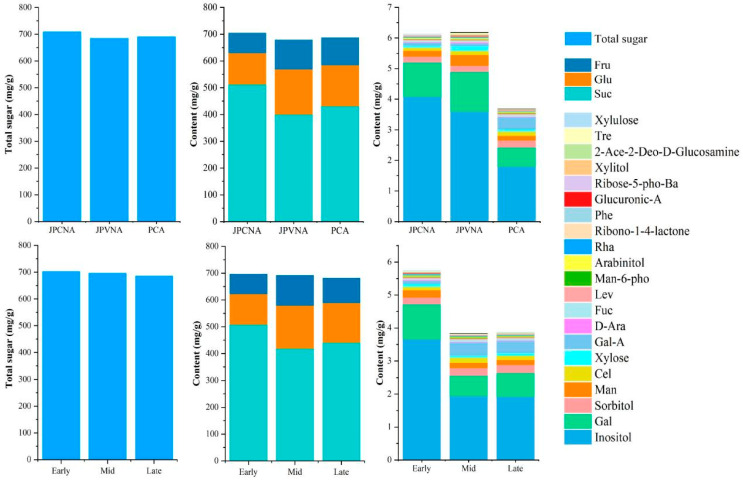
Stacked plots of soluble-sugar content of persimmon varieties of different types and from different sampling periods. (Sucrose, glucose and fructose contents were significantly higher than the other sugar components, so, in order to reflect the differences between the sugar components in each group more clearly, stacked-plot analyses were performed for the total sugar, the three main sugar components, and the residual sugar, respectively.).

**Figure 5 ijms-25-07803-f005:**
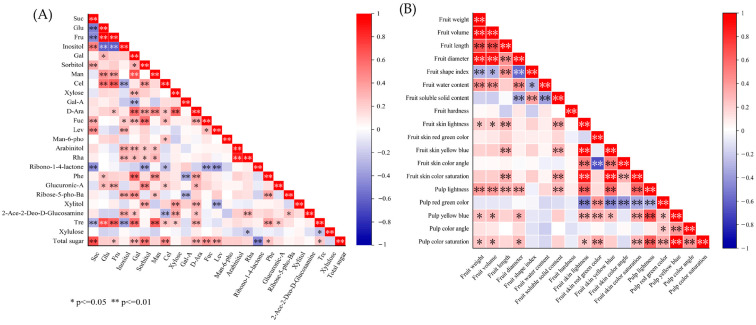
Correlation analysis of 25 sugar components (**A**) and 18 phenotypic traits (**B**) in 81 persimmon germplasm materials. *, and ** indicate significance at the 0.05 and 0.01 levels, respectively.

**Figure 6 ijms-25-07803-f006:**
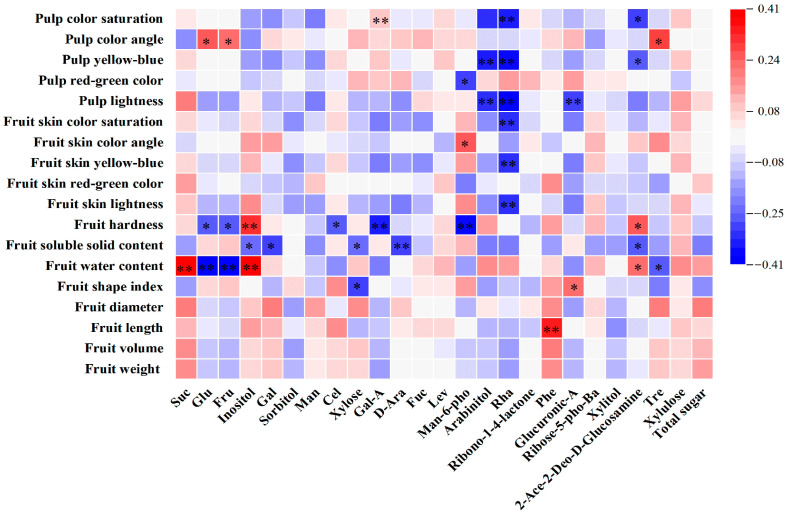
Heat map of correlations between soluble-sugar components and phenotypic traits in persimmon germplasm. * and ** indicate significance at the 0.05 and 0.01 levels, respectively.

**Figure 7 ijms-25-07803-f007:**
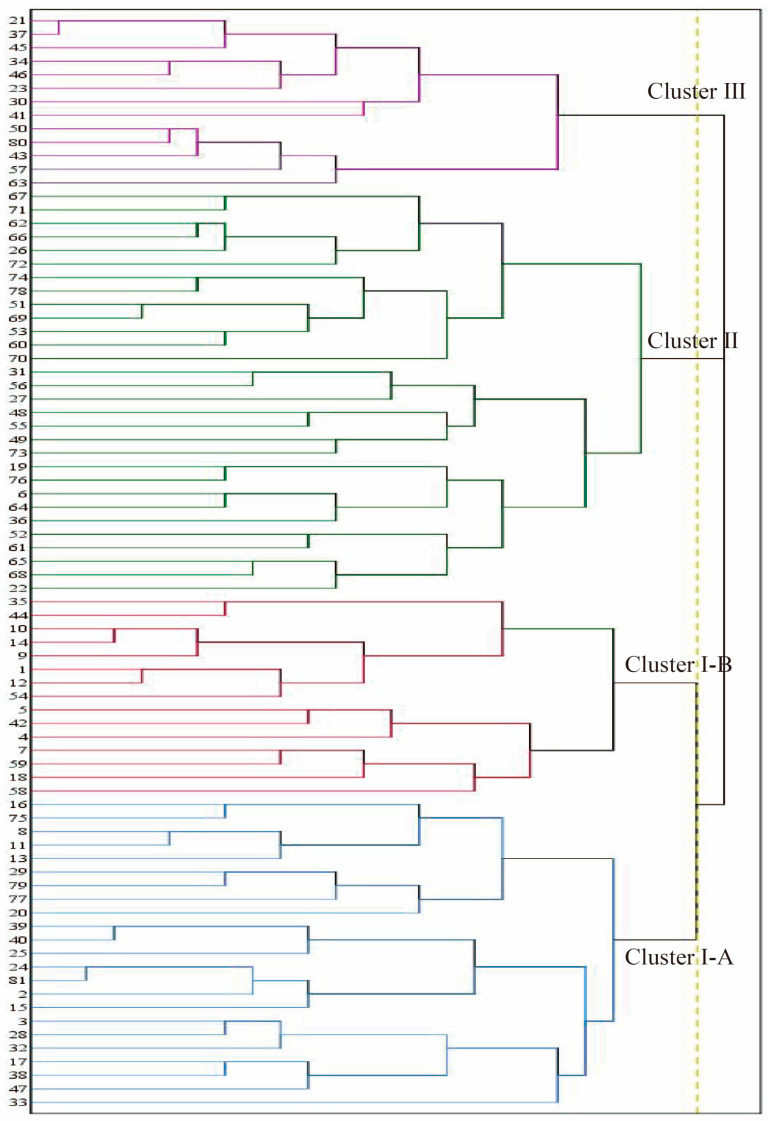
Persimmon germplasm clustering diagram.

**Figure 8 ijms-25-07803-f008:**
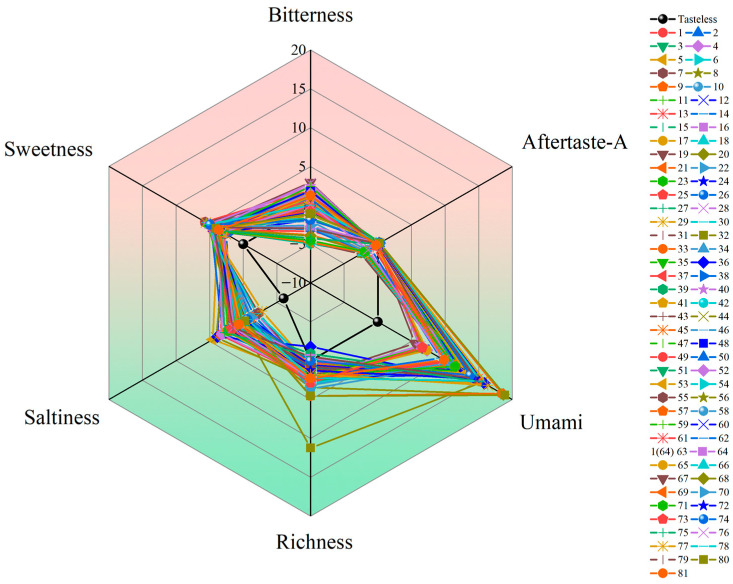
Radar chart of effective flavor indicators for persimmon germplasm. (The main sensory indicators for evaluating flavor are sourness, sweetness, astringency, umami, saltiness, bitterness, etc. Except for sourness (−13) and saltiness (−6), all other taste indicators had a tasteless point of 0).

**Figure 9 ijms-25-07803-f009:**
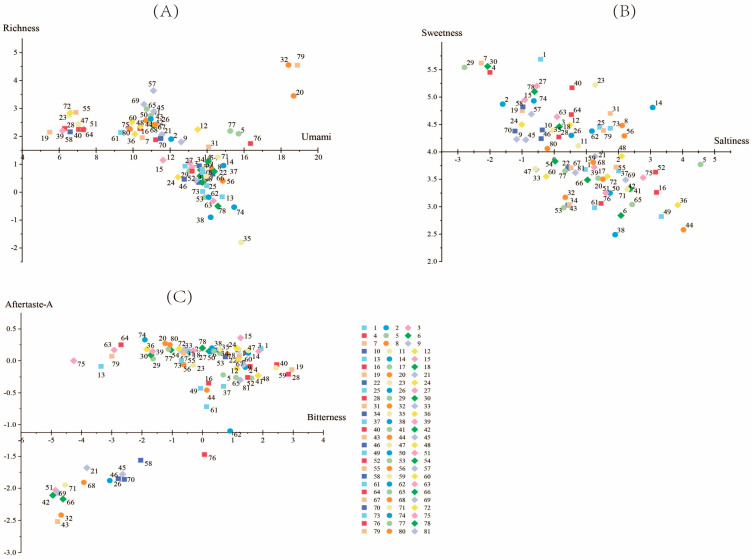
Scatterplot of persimmon germplasm. (**A**) for umami and richness; (**B**) for saltiness and sweetness; (**C**) for bitterness and aftertaste-A.

**Figure 10 ijms-25-07803-f010:**
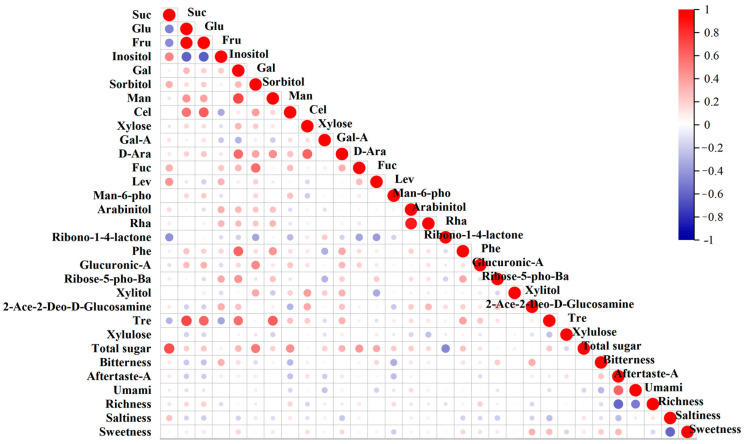
Correlation analysis between soluble-sugar components and taste in persimmon germplasm.

**Figure 11 ijms-25-07803-f011:**
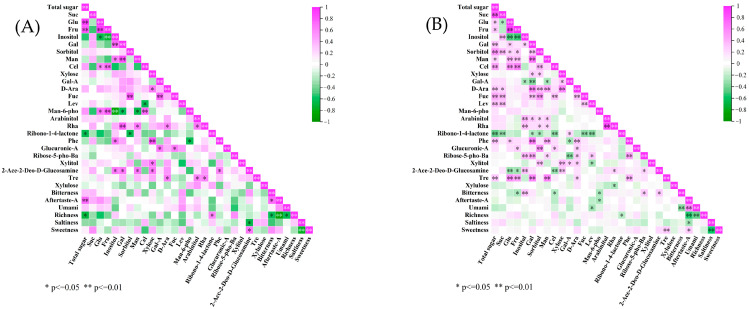
Analysis of correlation between soluble sugars and electronic-tongue data for persimmon germplasm. (**A**) shows the correlation between soluble sugars and electron-tongue data for hexose-accumulating persimmon germplasm; (**B**) shows the correlation between soluble sugars and electron-tongue data for sucrose-accumulating persimmon germplasm.

**Figure 12 ijms-25-07803-f012:**
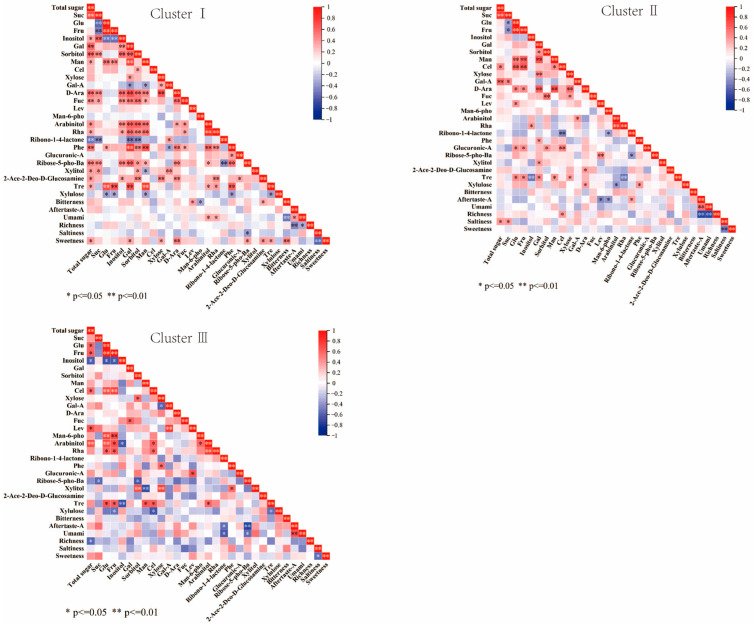
Persimmon germplasm soluble-sugar and electronic-tongue-data correlation analysis. From left to right are cluster I, cluster II and cluster III.

**Table 1 ijms-25-07803-t001:** Summary statistics of the 18 phenotypic traits studied in the 81 persimmon germplasm resources.

Traits	Mean	Max	Min	SD	R	CV%
Fruit weight (FW)/g	101.88	225.57	21.58	40.19	204.00	39.45%
Fruit volume (FV)/cm^3^	106.14	254.74	22.31	43.82	232.43	41.28%
Fruit length (FL)/cm	5.80	8.72	3.70	0.92	5.01	15.81%
Fruit diameter (FD)/cm	5.93	8.09	3.21	0.97	4.88	16.29%
Fruit shape index (FI)	0.99	1.47	0.76	0.17	0.71	17.44%
Fruit water content (FWC)/%	74.06	79.49	62.28	2.97	17.22	4.00%
Fruit soluble-solids content (FSSC)/Brix°	19.47	29.17	12.35	3.55	16.82	18.21%
Fruit hardness (FH)/kg/cm^3^	8.44	13.68	4.09	2.30	9.59	27.28%
Fruit skin lightness (SKL*)	64.10	70.84	49.20	3.69	21.64	5.76%
Fruit skin red-green color (Ska*)	19.00	29.93	6.05	5.00	23.88	26.33%
Fruit skin yellow-blue (SKb*)	61.36	70.50	37.34	5.60	33.15	9.13%
Fruit skin color angle (SKh)/°	71.25	78.41	30.44	6.13	47.97	8.60%
Fruit skin color saturation (SKc)	64.74	73.46	38.57	5.56	34.89	8.58%
Pulp lightness (PL*)	67.15	75.62	39.91	6.45	35.71	9.61%
Pulp red-green color (Pa*)	6.77	14.35	−1.28	3.44	15.63	50.79%
Pulp yellow-blue (Pb*)	40.07	52.59	15.57	7.04	37.02	17.56%
Pulp color angle (Ph)/°	68.57	84.62	−35.08	23.14	119.70	33.74%
Pulp color saturation (Pc)	40.88	53.79	15.99	7.06	37.80	17.28%

* R stands for extreme variance, i.e., max–min, and is often used to evaluate the dispersion of a set of data; CV is the coefficient of variation, which is the ratio of the standard deviation to the mean, expressed as a percentage (%).

**Table 2 ijms-25-07803-t002:** Summary statistics of 25 sugars in persimmon germplasm resources.

Component	Mean (mg/g)	Max (mg/g)	Min (mg/g)	SD	R (mg/g)	CV%
Total sugar	692.61	984.55	470.49	116.31	514.07	16.79%
Sucrose (Suc)	440	701.58	165.91	129.16	535.66	29.35%
Glucose (Glu)	149.46	379.45	64.42	58.06	315.02	38.84%
D-Fructose (Fru)	99.04	282.16	39.35	46.87	242.81	47.32%
Inositol	2.17	9.68	0.07	1.81	9.61	83.62%
D-Galactose (Gal)	0.72	2.72	0.15	0.44	2.57	61.08%
D-Galacturonic acid (Gal-A)	0.34	4.97	0	0.74	4.97	216.62%
D-Sorbitol	0.23	1.11	0.07	0.15	1.04	67.30%
D-Mannose (Man)	0.17	0.52	0.04	0.09	0.48	51.18%
Cellobiose (Cel)	0.13	0.28	0.08	0.04	0.2	29.87%
D-Xylose	0.08	0.23	0.03	0.04	0.21	56.11%
L-Fucose (Fuc)	0.06	0.18	0	0.04	0.18	65.98%
Levoglucosan (Lev)	0.04	0.06	0.02	0.01	0.05	29.96%
D-Arabinitol	0.03	0.06	0.02	0.01	0.04	25.59%
Trehalose (Tre)	0.03	0.17	0	0.03	0.17	97.98%
D-Mannose-6-phosphate sodium salt (Man-6-pho)	0.03	0.06	0.01	0.01	0.05	36.75%
L-Rhamnose (Rha)	0.02	0.03	0.01	0.46 × 10^−2^	0.02	23.44%
D-Arabinose (D-Ara)	0.02	0.04	0.01	0.01	0.03	30.48%
Phenylglucoside (Phe)	0.79 × 10^−3^	0.28 × 10^−2^	0	0.49 × 10^−3^	0.28 × 10^−2^	62.06%
2-Acetamido-2-deoxy-D-glucopyranose (2-Ace-2-Deo-D-Glucosamine)	0.22 × 10^−2^	0.14	0	0.02	0.14	710.44%
D-Ribono-1,4-lactone	0.02	0.05	0	0.02	0.05	77.50%
Barium D-ribose-5-phosphate (Ribose-5-pho-Ba)	0.01	0.05	0	0.01	0.05	107.32%
D-Glucuronic acid (Glucuronic-A)	0.01	0.06	0	0.01	0.06	103.80%
Xylitol	0.22 × 10^−2^	0.01	0	0.22 × 10^−2^	0.01	99.37%
Xylulose	0.12 × 10^−2^	0.45 × 10^−2^	0	0.10 × 10^−2^	0.45 × 10^−2^	88.42%

* R stands for extreme variance, i.e., max–min, and is often used to evaluate the dispersion of a set of data; CV is the coefficient of variation, which is the ratio of the standard deviation to the mean, expressed as a percentage (%).

**Table 3 ijms-25-07803-t003:** Soluble-sugar content of persimmon varieties of different types and from different sampling periods (mg/g).

Types	PCNA	PVNA	PCA	Early	Medium	Late
Total sugar	710.037 ± 120.937 ^a^	684.379 ± 74.812 ^a^	690.501 ± 118.719 ^a^	702.108 ± 83.915 ^a^	695.741 ± 128.351 ^a^	685.302 ± 112.958 ^a^
Suc	512.464 ± 149.919 ^a^	400.317 ± 115.404 ^a^	431.55 ± 124.845 ^a^	508.9 ± 130.845 ^a^	419.374 ± 138.942 ^a^	441.493 ± 109.674 ^a^
Glu	117.061 ± 44.854 ^a^	168.694 ± 75.126 ^a^	153.15 ± 57.962 ^a^	114.177 ± 44.198 ^a^	160.746 ± 63.64 ^a^	147.787 ± 50.818 ^a^
Fru	74.383 ± 26.606 ^a^	109.185 ± 53.162 ^a^	102.119 ± 48.267 ^a^	73.296 ± 28.087 ^b^	111.782 ± 55.053 ^a^	92.153 ± 35.393 ^ab^
Inositol	4.084 ± 2.935 ^a^	3.596 ± 2.099 ^a^	1.799 ± 1.34 ^a^	3.661 ± 2.633 ^a^	1.942 ± 1.536 ^b^	1.927 ± 1.592 ^b^
Gal	1.122 ± 0.855 ^a b^	1.305 ± 0.212 ^a^	0.623 ± 0.277 ^b^	1.062 ± 0.625 ^a^	0.623 ± 0.245 ^a^	0.716 ± 0.502 ^a^
Sorbitol	0.191 ± 0.073 ^a^	0.198 ± 0.047 ^a^	0.238 ± 0.167 ^a^	0.206 ± 0.073 ^a^	0.224 ± 0.141 ^a^	0.247 ± 0.191 ^a^
Man	0.194 ± 0.115 ^a^	0.36 ± 0.127 ^a^	0.152 ± 0.061 ^a^	0.226 ± 0.136 ^a^	0.164 ± 0.067 ^a^	0.15 ± 0.077 ^a^
Cel	0.097 ± 0.019 ^b^	0.123 ± 0.031 ^ab^	0.139 ± 0.04 ^a^	0.097 ± 0.014 ^b^	0.148 ± 0.042 ^a^	0.126 ± 0.032 ^b^
Xylose	0.077 ± 0.023 ^a^	0.163 ± 0.082 ^a^	0.075 ± 0.04 ^a^	0.092 ± 0.052 ^a^	0.075 ± 0.037 ^a^	0.082 ± 0.051 ^a^
Gal-A	0.092 ± 0.13 ^a^	0.089 ± 0.08 ^a^	0.393 ± 0.802 ^a^	0.092 ± 0.107 ^a^	0.393 ± 0.877 ^a^	0.364 ± 0.673 ^a^
D-Ara	0.019 ± 0.005 ^b^	0.028 ± 0.009 ^a^	0.018 ± 0.005 ^b^	0.021 ± 0.008 ^a^	0.019 ± 0.005 ^a^	0.017 ± 0.005 ^a^
Fuc	0.051 ± 0.033 ^a^	0.052 ± 0.01 ^a^	0.062 ± 0.042 ^a^	0.05 ± 0.018 ^a^	0.065 ± 0.043 ^a^	0.058 ± 0.041 ^a^
Lev	0.038 ± 0.009 ^a^	0.036 ± 0.006 ^a^	0.039 ± 0.012 ^a^	0.036 ± 0.007 ^a^	0.038 ± 0.012 ^a^	0.039 ± 0.012 ^a^
Man-6-pho	0.017 ± 0.005 ^b^	0.015 ± 0.003 ^b^	0.027 ± 0.009 ^a^	0.02 ± 0.007 ^b^	0.029 ± 0.01 ^a^	0.022 ± 0.007 ^b^
Arabinitol	0.041 ± 0.012 ^a^	0.043 ± 0.01 ^a^	0.032 ± 0.007 ^a^	0.042 ± 0.01 ^a^	0.031 ± 0.008 ^b^	0.034 ± 0.007 ^b^
Rha	0.023 ± 0.006 ^a^	0.025 ± 0.004 ^a^	0.019 ± 0.004 ^a^	0.024 ± 0.004 ^a^	0.019 ± 0.005 ^b^	0.019 ± 0.004 ^b^
Phe	0.001 ± 0.001 ^a^	0.001 ± 0.001 ^a^	0.001 ± 0 ^a^	0.001 ± 0.001 ^a^	0.001 ± 0 ^a^	0.001 ± 0 ^a^
Tre	0.025 ± 0.022 ^a^	0.051 ± 0.029 ^a^	0.026 ± 0.026 ^a^	0.026 ± 0.02 ^a^	0.028 ± 0.03 ^a^	0.026 ± 0.023 ^a^
2-Ace-2-Deo-D-Glucosamine	0.002 ± 0.004 ^a^	0.037 ± 0.069 ^a^	0 ± 0 ^a^	0.015 ± 0.042 ^a^	0 ± 0 ^a^	0 ± 0.001 ^a^
Ribono-1-4-lactone	0.022 ± 0.012 ^a^	0.023 ± 0.017 ^a^	0.02 ± 0.016 ^a^	0.027 ± 0.01 ^a^	0.018 ± 0.017 ^a^	0.022 ± 0.015 ^a^
Ribose-5-pho-Ba	0.02 ± 0.02 ^a^	0.01 ± 0 ^a^	0.01 ± 0.01 ^a^	0.02 ± 0.01 ^a^	0.01 ± 0.01 ^a^	0.01 ± 0.01 ^a^
Glucuronic-A	0.02 ± 0.017 ^a^	0.008 ± 0.003 ^a^	0.009 ± 0.01 ^a^	0.016 ± 0.014 ^a^	0.011 ± 0.01 ^a^	0.008 ± 0.012 ^a^
Xylitol	0.002 ± 0.002 ^a^	0.002 ± 0.001 ^a^	0.002 ± 0.002 ^a^	0.002 ± 0.002 ^a^	0.002 ± 0.002 ^a^	0.002 ± 0.002 ^a^
Xylulose	0.001 ± 0.002 ^a^	0.001 ± 0.001 ^a^	0.001 ± 0.001 ^a^	0.001 ± 0.001 ^a^	0.001 ± 0.001 ^a^	0.001 ± 0.001 ^a^

* PCNA, pollination-constant non-astringent; PVNA, pollination-variant non-astringent; PCA, pollination-constant astringent; early, varieties with short growing days; medium, varieties with moderate growing days; late, varieties with long growing days. Different lowercase letters indicate significant differences among the treatments (Duncan’s multiple range test, *p* < 0.05).

**Table 4 ijms-25-07803-t004:** Eigenvalues and contribution rates of principal components.

PCA	Eigenvalues	Variance Contribution Rate (%)	Cumulative Variance Contribution Rate (%)
1	4.918	20.493	20.493
2	4.112	17.133	37.625
3	2.424	10.100	47.725
4	2.033	8.470	56.195
5	1.704	7.100	63.295
6	1.347	5.613	68.908
7	1.125	4.689	73.597

**Table 5 ijms-25-07803-t005:** The component matrix after the principal component has been rotated on each sugar-component index.

Sugar Component Indicators	PC1	PC2	PC3	PC4	PC5	PC6	PC7
Man	0.732	0.394	−0.190	−0.111	−0.110	−0.166	0.003
Tre	0.725	−0.275	−0.218	−0.270	−0.110	0.122	−0.032
Glu	0.707	−0.556	0.041	−0.325	−0.056	−0.062	−0.037
D-Ara	0.681	0.149	0.163	0.400	−0.280	−0.057	0.071
Fru	0.672	−0.630	0.069	−0.279	0.013	−0.009	0.005
Phe	0.602	0.451	−0.076	0.057	−0.301	0.100	−0.088
Xylose	0.546	−0.067	−0.134	0.530	−0.130	0.135	0.018
Glucuronic-A	0.486	0.125	0.048	0.263	0.069	−0.355	0.399
2-Ace-2-Deo-D-Glucosamine	0.480	0.230	−0.312	0.300	−0.117	−0.354	0.283
Xylulose	−0.346	0.000	−0.040	0.241	−0.313	0.048	0.108
Inositol	−0.023	0.836	0.098	−0.020	0.098	0.173	−0.081
Gal	0.588	0.603	−0.020	−0.094	−0.128	0.168	−0.280
Arabinitol	0.337	0.599	−0.042	−0.169	0.561	0.111	0.190
Ribose-5-pho-Ba	0.171	0.551	0.030	−0.336	−0.324	0.328	−0.274
Suc	−0.282	0.545	0.535	0.097	0.071	0.003	0.006
Fuc	0.190	0.145	0.737	0.105	−0.015	−0.114	−0.173
Sorbitol	0.330	−0.162	0.620	0.395	0.285	0.152	−0.135
Ribono-1-4-lactone	0.073	−0.142	−0.557	0.192	0.446	−0.078	−0.423
Cel	0.460	−0.471	0.478	−0.255	−0.040	0.175	0.040
Xylitol	0.130	−0.307	0.131	0.616	0.135	0.570	0.159
Man-6-pho	0.013	−0.067	0.281	−0.511	0.099	0.210	0.495
Rha	0.375	0.498	−0.089	−0.131	0.645	−0.015	0.154
Lev	−0.191	0.333	0.447	−0.121	−0.257	−0.484	0.008
Gal-A	0.267	−0.329	0.336	0.047	0.362	−0.402	−0.374

**Table 6 ijms-25-07803-t006:** Specific information on 81 persimmon germplasm resources.

No.	Name	Type	Sampling	No.	Name	Type	Sampling
1	Jiro	JPCNA	2022/10/27	42	Dahongpaoshi	PCA	2022/11/9
2	Haze-gosho	JPCNA	2022/11/9	43	lvnaitou	PCA	2022/11/3
3	Suruga	JPCNA	2022/11/9	44	YL 250	PCA	2022/11/3
4	Touyouichi	JPVNA	2022/10/27	45	YL 261	PCA	2022/11/3
5	Shougatsu	JPVNA	202210/27	46	YL 264	PCA	2022/11/9
6	Oku-gosho	JPCNA	2022/10/27	47	YL 265	PCA	2022/11/9
7	Yamafuji	JPVNA	2022/11/9	48	YL 272	PCA	2022/11/9
8	Kazusa	JPCNA	2022/10/27	49	YL 274	PCA	2022/11/9
9	Sinami	JPCNA	2022/10/27	50	YL 275	PCA	2022/11/9
10	T	JPCNA	2022/10/27	51	YL 277	PCA	2022/11/3
11	Z	JPCNA	2022/11/9	52	YL 286	PCA	2022/11/3
12	Gosho	JPCNA	2022/11/9	53	YL 288	PCA	2022/11/3
13	Niuyanshi	PCA	2022/11/9	54	YL 291	PCA	2022/11/9
14	Fuyu	JPCNA	2022/11/9	55	YL 293	PCA	2022/11/9
15	H	JPCNA	2022/10/27	56	YL 296	PCA	2022/11/3
16	Hongyushi	PCA	2022/11/9	57	YL 301	PCA	2022/11/9
17	Anxiniuxinshi	PCA	2022/11/3	58	YL 304	PCA	2022/11/3
18	YangshuoNiuxinshi	PCA	2022/11/9	59	YL 317	PCA	2022/11/3
19	Guangzhoudaniuxinshi	PCA	2022/11/3	60	YL 323	PCA	2022/11/3
20	Yongshuntezaoshi	PCA	2022/11/3	61	YL 325	PCA	2022/11/3
21	Haianxiaofangshi	PCA	2022/11/3	62	YL 326	PCA	2022/11/3
22	Bianshi	PCA	2022/10/27	63	YL 330	PCA	2022/11/9
23	Jinhongshi	PCA	2022/11/3	64	YL 332	PCA	2022/11/9
24	Dashuishi	PCA	2022/11/3	65	YL 334	PCA	2022/11/9
25	Niunaishi	PCA	2022/11/9	66	YL 336	PCA	2022/11/3
26	Xiaobaxianshi	PCA	2022/11/9	67	YL 337	PCA	2022/11/3
27	Gongchengyueshenshi	PCA	2022/11/9	68	YL 338	PCA	2022/11/9
28	Anxiyoushi	PCA	2022/11/3	69	YL 339	PCA	2022/11/3
29	Houzishi	PCA	2022/11/9	70	YL 340	PCA	2022/11/3
30	Boaibayuehuang	PCA	2022/11/9	71	YL 341	PCA	2022/11/3
31	Chengjiangniuxinshi	PCA	2022/11/9	72	YL 344	PCA	2022/11/3
32	Guangzhou jixinshi	PCA	2022/11/9	73	YL 348	PCA	2022/11/3
33	YL 197	PCA	2022/11/3	74	YL 349	PCA	2022/11/9
34	YL 200	PCA	2022/11/3	75	YL 358	PCA	2022/11/3
35	YL 201	PCA	2022/11/3	76	YL 359	PCA	2022/11/9
36	YL 202	PCA	2022/11/3	77	YL 362	PCA	2022/11/3
37	YL 204	PCA	2022/11/3	78	YL 363	PCA	2022/11/3
38	YL 211	PCA	2022/10/27	79	Taiwanhongshi	PCA	2022/11/3
39	Jiangxiseshi	PCA	2022/10/27	80	YL 711	PCA	2022/11/3
40	Qiaodingshi	PCA	2022/11/3	81	Fangshi	PCA	2022/11/3
41	Weiboshi	PCA	2022/11/3				

## Data Availability

The data presented in this study are available upon request from the corresponding author. The data are not publicly available due to privacy reasons.

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
