# Peer review of "Analysis of the Correlation between Persimmon Fruit-Sugar Components and Taste Traits from Germplasm Evaluation"

_ijms, 2024, doi:10.3390/ijms25147803_

Round 1

Reviewer 1 Report

Comments and Suggestions for Authors

General remarks: the term “germplasm” is used which seems to be inadequate in the context of this report; the aim of the study is unclear, the title suggests investigations on molecular level which was not done.

Specific comments:

1)     Abstract: requires substantial improving, it is unclear what, for what purpose and how was investigated;

2)     All Latin names of plant genera and species have to be written using Italic font;

3)     “Glew [27] and Ge [28]” – proper citation style is required; and so on;

4)     Line 79: one sugar?

5)     Line 97: it is highly unclear why the term: “germplasm” is used, nothing indicates that genetic resources are investigating in this study;

6)     The aim of the study should be clearly stated;

7)     It is not clearly stated what kind of plant material was investigated, the repeated term: ”germplasm” is highly confusing as it indicates genetic resources collected for protecting bioresources;

8)     In Discussion section it is stated: “The study of the genetic diversity of phenotypic traits..” – but no genetic studies were performed.

Reviewer 2 Report

Comments and Suggestions for Authors

The publication precisely and clearly shows the differentiation of cultivars in terms of sugar production, which will be helpful in the selection of varieties.

Good work!

Comments on the Quality of English Language

Genus and species names should always be italicized.

The most recent edition of the International Code of Nomenclature for algae, fungi, and plants recommends that all plant names be in a different font from the rest of the text. The Royal Horticultural Society (U.K.) recommends that family names be italicized.

Author Response

Thank you for your comments concerning our manuscript. Those comments are all valuable and very helpful for revising and improving our paper, as well as the important guiding significance to our researches. Thank you for your support and affirmation of this study, and we will certainly take the subsequent improvement work seriously.

Reviewer 3 Report

Comments and Suggestions for Authors

The manuscript proposes a careful study of persimmon germplasm as a function of phenotypic traits, sugar components and taste variables. The number of persimmon germplasms is very high and this gives added value to the study.

Minor changes are suggested:

Line 19: Explain what PCNA and PCA are.

Table 1 and paragraph 2.1: It would be appropriate to indicate the color coordinates of the pulp and skin as L*, a*, b*, etc.

Line 265: There is no figure S4.

Paragraph 2.4: Considering the variance contribution rate for each PC, all 7 PCs must be explained and not just the first 2.

Tables 4-5: Uniform the number of significant figures for each variable.

Paragraph 2.5: In practice, there are 10 main clusters and not 3. The division into just 3 macro-clusters is reductive. However, the 3 macro-clusters can be considered subsequently, but in this paragraph all 10 must be considered and explained.

Figure 5: The subdivision of clusters I, II, III and the related legend are missing.

Paragraph 2.8: This paragraph is useless, considering the very low R2 values, and therefore should be eliminated.

Paragraph 3.4: Since the regression results are not significant, the paragraph should be eliminated.

Line 648: How was commercial maturity determined? According to what was written subsequently, there was not an objective evaluation through instrumentation, but a subjective one by the operator. This may have introduced greater variability.

Line 652: How were the fruits chosen for harvesting?

Line 672: indicate the freeze dryer model and the size of the powder particles.

Comments on the Quality of English Language

Minor editing of English language required

Round 2

Reviewer 1 Report

Comments and Suggestions for Authors

None.
